# Sorption of Sulfamethoxazole on Inorganic Acid Solution-Etched Biochar Derived from Alfalfa

**DOI:** 10.3390/ma14041033

**Published:** 2021-02-22

**Authors:** Qi Li, Wei Yu, Linwen Guo, Yuhang Wang, Siyu Zhao, Li Zhou, Xiaohui Jiang

**Affiliations:** 1College of Urban and Environmental Sciences, Northwest University, Xi’an 710127, China; yuwei@stumail.nwu.edu.cn (W.Y.); yuhangwang895@stumail.nwu.edu.cn (Y.W.); 201931906@stumail.nwu.edu.cn (S.Z.); zhouli233@stumail.nwu.edu.cn (L.Z.); xhjiang@nwu.edu.cn (X.J.); 2Ningdong Forestry Bureau of Shaanxi Province, Xi’an 710127, China; qili726@163.com

**Keywords:** biochar, etch, sulfamethoxazole, adsorption, mechanism

## Abstract

The properties of alfalfa-derived biochars etched with phosphoric (PBC) or hydrochloric acid (ClBC) compared with raw materials (BC) were examine in this paper. SEM, FT-IR, XRD, BET and elemental analysis were performed to characterize the micromorphology and chemical structure comprehensibly. The results showed that the porous structure was enhanced, and surface area was increased via etching with inorganic acids. Batch adsorption experiments were performed for sulfamethoxazole (SMX) to biochars. The experimental data showed that modified biochars exhibited higher adsorption capacity for SMX, i.e., the adsorption quantity of ClBC and PBC had risen by 38% and 46%. The impact on pH values suggested that the physisorption, including pore-filling and electrostatic interaction, might be applied to original biochar. In addition, chemisorption also played a role, including hydrogen bonding, π-π electron donor acceptor interaction (π-π EDA), and so on. Furthermore, both pH and coexisting ions also had a certain effect on sorption. Enhancement of the electrostatic attraction between biochar and SMX might also account for the enhanced capacity of SMX at pH < 7, and coexisting ions could decrease the amount of SMX adsorbed onto biochars, mainly because of competition for adsorption sites.

## 1. Introduction

At present, antibiotics have been detected in aquatic environments. Subsequently, the environmental hazards and public health problems associated with antibiotics are arousing great concern [1]. According to a survey, the total use of antibiotics is greater than 92,700 tons per year, with an estimated human and animal discharge of more than 54,000 tons of antibiotics, and ultimately more than 53,800 tons of antibiotics entering the receiving environment via various sewage treatments [2]. The overuse of antibiotics has caused serious environmental problems. The release of antibiotics has led to the spread of antibiotic resistance. Additionally, antibiotics are a type of chemical pollutant that pose a threat to human health by affecting the retention and transmission of resistance genes in ecosystems [3,4,5]. Although there is little risk of direct toxicity to species at current concentrations of antibiotics in the environment, indirect effects induced by antibiotics may alter the abundance of ecological species, thus altering community composition, affecting human dysfunction and causing fetal malformation [6]. Therefore, antibiotics pose serious health risks to the ecosystem.

Sulfamethoxazole (SMX) is a common antibiotic that is widely used around the world [7]. Because of its cost–benefit advantages and broad-spectrum bactericidal effect, SMX is widely used in human and animal drugs to kill bacteria and prevent the spread of disease [8]. As sulfamethoxazole starts to move into large-scale usage, antibiotic products and their metabolites will inevitably enter into the aquatic environment. Neutral forms of SMX have pH values predominantly between 1.8 and 5.6, while retaining a high degree of hydrophilicity (octanol water partition coefficient log K_ow_ = 0.89) [9]. SMX can be present in water in several forms, including as uncharged molecules (SMX^0^), zwitterions (SMX^±^), cations (SMX^+^) and anions (SMX^−^), owing to proton exchange between the aromatic amine and sulfonamide groups on the molecule (pKa_1_,1.8; pKa_2_, 5.6) [10]. In aqueous solution, the adsorption behavior of SMX during removal is complicated due to its complex morphology. Currently, it is difficult to achieve satisfactory results using conventional biological treatment processes.

Nowadays, the technologies for the removal of sulfamethoxazole in water environments include adsorption [11], biodegradation [12] and chemical methods [13]. Among these, adsorption has the advantages of simple operation and high removal efficiency, and is presently the most commonly used removal method [14]. Common adsorbents include carbon nanotubes [15], graphene oxide [16], clay minerals [17], activated carbon [18], and biochars. The most commonly used feedstocks for producing biochar include biomass and minerals, such as crop straw [19], wood chips [20], peels [21], sludge [22] and animal manure [23]. The surface morphology, pore structure, pH value, aromaticity, hydrophilicity and polarity of biochar obtained by pyrolysis from different raw materials are significantly different. To design new materials with higher selective properties and strong adsorption capacity, a variety of modified biochars have been produced and have been widely used to remove antibiotics in wastewater [24]. Since the physicochemical properties of the original biochar have not fully been explored, a variety of chemical and physical means of modification have been proposed to further develop the potential of pore and functional groups [25,26]. Chemical methods have been broadly used to improve the surface physical and chemical properties of biochar. Through modification, the physical and chemical properties of biochar, including hydrophobicity and aromaticity, specific surface area, and porous structure, can be improved to a certain extent [27]. Numerous studies have reported the modification of biochar using a single acidic or alkaline solution. However, comparative studies on the mechanism of biochar modified with acid solution for the removal of antibiotics have rarely been reported. Compared with strong oxidizing acids, hydrochloric acid and phosphoric acid are two common and cheap inorganic acids, and the properties of biochar can be improved to a certain extent through chemical etching.

The adsorption capacity of biochar depends on the modification method; additionally, the source and type of biomass is a crucial factor. Therefore, the selection of effective and sustainable raw materials is crucial [28]. The adsorption of sulfonamides by natural soil, clay minerals and humus has been reported many times in previous studies [29]. Previous studies have shown that the main mechanisms driving adsorption are hydrophobicity, hydrogen bonding and surface complexation. A suitable selection of biomass with physicochemical properties is key. Alfalfa was one of the most abundant cultivated herbages in the world, possessing high yield and good quality. It has been cultivated in China for more than 2000 years. In addition, it has been reported in relevant literature that alfalfa is an excellent source of animal feed due to its rich content of protein, minerals and cellulose [30]. High levels of protein content increase the content of nitrogen in alfalfa biochar, leading to the formation of more amino groups on the surface. Additionally, the framework of cellulose forms a porous structure after pyrolysis at high temperature. The formation of the ordered carbon structure contributes greatly to the new properties of biochar. The presence of endogenous minerals has a significant catalytic effect on the pyrolysis of biomass, and greatly reduces the decomposition temperature of the carbon skeleton [31]. Alfalfa could be selected as a potentially suitable raw material.

The goal of this study is to investigate the absorption mechanism and influencing factors of sulfamethoxazole using a new type of hay-based biochar in simulated wastewater. Sulfamethoxazole, as one of the most frequently prescribed antiphlogistic agents belonging to the sulfonamides, was selected as an experimental compound. Biochar derived from alfalfa was pyrolyzed at 800 °C and then etched with hydrochloric acid or phosphoric acid, which were used as a model adsorbent and modified adsorbent. The aims of this work are as follows. Firstly, to examine the compositions and structures of alfalfa biochars prepared using different methods. Secondly, to illustrate the mechanisms operating between the typical antibiotic sulfamethoxazole and three biochars via sorption kinetics, isotherms and thermodynamics. Apart from the above, different impacts, including initial solution pH, ion types and strength, are investigated and optimized.

## 2. Materials and Methods

### 2.1. Reagents and Biomass

Alfalfa (AF) was collected from the Baofa Agriculture and Animal Husbandry Company of Guyuan in Ningxia, China. SMX (analytical grade, purity >99%) were from Shanghai source biological technology Co. LTD, Shanghai, China and used without further purification. All reagents in this work were of analytical grade purity and above and were bought from Comiou Chemical Reagent Co. LTD, Tianjin, China. All ultra-pure water used in the experiments had a resistivity of 18.2 MΩ.

### 2.2. Modification of BCs (Sorbents)

The alfalfa specimens were ground (100 mesh), soaked and cleaned with ultra-pure water and then thoroughly oven dried at 80 °C for 2 days. After that, it was pyrolyzed under a nitrogen atmosphere at 800 °C for 2 h. The obtained matter was cleaned with ultra-pure water and dried at 80 °C for 48 h in an oven; this was designated as BC. The raw biochar was modified by specified acids, this process can be described in detail as follows: 10 g primary biochar was placed into two beakers, one was combined with 50 mL 14% H_3_PO_4_ solution; to the other was added 50 mL 1 mol·L^−1^ HCl solution. Both of them were soaked for 24 h and then rinsed until the pH of the eluent liquid was around 7.0. These three sorbents were dried at 80 °C for 48 h. After cooling to room temperature, they were stored in reagent bottles for later use. The BC modified by H_3_PO_4_ was designated as PBC, and that modified by HCl was designated as ClBC.

### 2.3. Characterization of BCs

The total pore volumes and pore diameters of the BCs were evaluated using the Brunauer-Emmett-Teller (Quantachrome Instruments, NOVA 4200, Boynton Beach, FL, USA). The surface structure and morphology of the AF-BCs were analyzed using scanning electron microscopy (FEI Quenta 400 FEG, Hillsboro, OR, USA). The chemical properties of BCs were highlighted by a Fourier transform infrared spectroscopy analysis (Nicolet Instruments, Nexus 870, Madison, WI, USA). The carbon, hydrogen, oxygen and nitrogen analysis were performed on an element analysis instrument (EuroVector, EA3000, Pavia, Italy). The aromaticity and polarity of the material can be evaluated by atomic ratio (H/C and (O + N)/C). XRD was applied to investigate the crystalline structure of these biochars (Rigaku Corporation, Rigaku MiniFlex II, Tokyo, Japan). Moreover, pH_pzc_ was tested using the pH drift method [32].

### 2.4. Batch Adsorption Experiment

In order to determine the adsorption isotherm of SMX on BCs, batch experiments were conducted in 100 mL brown conical flasks at 288.15 K, 298.15 K and 308.15 K, respectively. Sulfamethoxazole simulating contaminants was dissolved in ultra-pure water containing 2% methanol. A total of 50 mg of the adsorbent was mixed with 100 mL of sulfamethoxazole aqueous solutions (a series of initial concentration from 10 to 60 mg·L^−1^) was shaken at 298.15 K for 72 h until adsorption equilibrium was reached. Batch adsorption experiments were conducted to evaluate the effects of other factors on the reaction. To determine the impact of pH, the initial pH was adjusted to various values using HCl and NaOH. For the impact of ionic types on adsorption experiments, 0.1 mol·L^−1^ NaCl, KCl, MgCl_2_, AlCl_3_, NaNO_3_, NaHCO_3_, Na_2_CO_3_, Na_2_SO_4_ were added to the SMX solutions. The concentration of NaCl from 0.05 to 1.0 mol·L^−1^ was applied on the effects of ions experiments. Each of these experiments was performed in triplicate, and the data were averaged. The solutions were filtered using a 0.22 μm syringe filter and detected by UV spectrophotometer (Shimadzu Corporation, UV-1800, Tokyo, Japan) at 265 nm.

### 2.5. Data Analysis

The adsorption data was fitted to three kinetic equations, respectively: pseudo-first order (PFO), pseudo-second order (PSO) and intra-partical-diffusion (IPD) [33]. The equations employed in the study are provided below:(1)$$\mathrm{PFO}:\;Q_{t}=Q_{e}(1-\exp(-K_{1}t))$$
(2)$$\mathrm{PSO}:\;Q_{t}=K_{2}Q_{e}{}^{2}t/(1+K_{2}Q_{e}t)$$
(3)$$\mathrm{IPD}:\;Q_{t}=K_{i}t^{\frac{1}{2}}+C$$
where *K*_1_, *K*_2_ and *K_i_* are the rate constants, respectively. *C* is the sorption constant. *Q_t_* and *Q_e_* are the adsorption quantity of biochar.

The data for the sorption isotherms were examined using three models to check the reaction behavior between SMX and biochar. The equations were as follows:(4)$$\mathrm{Freundlich}:\;q_{e}=K_{F}C_{e}{}^{\frac{1}{n}}$$
(5)$$\mathrm{Langmuir}:\;q_{e}=q_{\max}K_{L}C_{e}/(1+K_{L}C_{e})$$
(6)$$\mathrm{Temkin}:\;q_{e}=\frac{RT}{b}\ln(AC_{e})$$
where *q*_max_ and *C_e_* are the maximum capacity and equilibrium concentration. *K_F_* and *K_L_* are the Freundlich and Langmuir constant, and n is heterogeneity. A and B are the Temkin isotherm constant.

In addition, the calculation of thermodynamic parameters was performed using Van’t Hoff equations as follows:(7)ΔG=−RTlnK
(8)$$\ln K=-\frac{\Delta H}{RT}+\mathrm{constant}$$
(9)$$\Delta S=\frac{\Delta H-\Delta G}{T}$$
where *R* is a constant of 8.314 (KJ·mol) ^−1^, *T* is the Kelvin temperature, and *K* is the equilibrium dissociation constant.

## 3. Results and Discussion

### 3.1. Characterization

Scanning electron microscope (SEM) images of BC, ClBC and PBC were shown in Figure 1. Evenly distributed and tangled porous micro-tube-like structures can be observed. Notably, there were many small pores in the pore wall of these tubes. It is known that organic matter such as cellulose hemicellulose and lignin breaks down during pyrolysis at high temperatures, leaving behind porous structures. The surface of modified biochar was smoother than that of raw carbon. A possible reason for this was that the impurities in the pores of biochar, such as calcium carbonate and magnesium carbonate after acid impregnation, were reduced. On the one hand, he abundant porous structure improved the specific surface area of the biochars; on the other hand, it also promoted the diffusion of ions on the surface.

Similarly, as shown in Appendix A (refer to Appendix A, the specific surface areas of ClBC and PBC were higher than those of the raw ones. The specific surface area of biochar eroded by acid solution was slightly increased. The likely reason for this is that the acid solution washes organic compounds off the biochar’s surface [7]. It was also found that biochars etched by acid possessed larger pore volumes. Interestingly, the pore size of biochar increased with the addition of phosphate solution. From Appendix A (refer to Appendix A), it showed that the content of ash and oxygen was less on ClBC than on others. In contrast, the ClBC subsumed more carbon and nitrogen than others. Thus, it can be concluded that biochar treated with acid had significant changes in terms of structure and element content. Specifically, the biochars with modifications exhibited lower polarity index ((N + O)/C) and hydrophilicity (O/C) than the raw ones. The main cause was the acid treatment enabling more removal of inorganic matter, and the concomitant increase in the proportion of fixed carbon. This was also a major factor leading to a slight increase in the aromaticity (H/C) of acidic biochars.

XRD can be used to analyze the phase composition and crystal structure of biochar. The diffraction peak of amorphous crystal was weak, and was a steamed bread peak or even no peak, while the diffraction peak of crystal was sharp. From Figure 2A, the XRD spectra of BC and ClBC exhibit two weak and broad diffraction peaks at 16.53° and 29.95°. In addition, the XRD pattern of PBC showed only one weak and broad diffraction peak at 25°. It turned out that the formulated materials provide evidence that all biochars are amorphous. Particularly, there are strong and sharp peaks at 20.48° and 26.35° for ClBC and PBC, respectively, which indicates that some peaks belonging to impurities were being observed simultaneously. The impurities were determined to be calcium carbonate and silicon dioxide through comparison with standard PDF cards. The results showed that the structure of biochar was more complete after etching by acid, which was also conducive to the diffusion of pollutants on its surface.

To determine the corresponding stretching and vibration of different molecular bonds, the biochars were analyzed by FT-IR. The spectra of BC, ClBC and PBC are shown in Figure 3. The spectra of all samples showed a broad band at 400–4500 cm^−1^, which was attributed to aromatic compounds (670 cm^−1^, 790 cm^−1^ and 875 cm^−1^), and vibration of alcoholic groups (C-O stretching, 1076 cm^−1^), alkanes -CH_2_ ring stretching (1434 cm^−1^), aromatic C=C and C=N ring stretching (1556 cm^−1^), C=O stretching of carboxyl (1716 cm^−1^), C≡C and C≡N stretching alkynes (2343 cm^−1^ and 2364 cm^−1^), -OH stretching of alcoholic and phenolic (3654 cm^−1^). By comparison, the vibrations of the carboxyl and hydroxyl molecular bonds on the etched biochar were not obvious, but alkoxy increased. The results suggested that oxygen-containing functional groups increased after acid solution modification [34]. With the reduction of the vibration intensity of the alkyne, the sample modified by phosphate had stronger polarity, but the biochar modified with hydrochloric acid was the opposite.

### 3.2. Adsorption Kinetics

To further study the limiting steps and the mechanism of adsorption rate, the kinetics were explored in this study [35]. As shown in Figure 3, for SMX, there are two main factors affecting the adsorption amount: one is the reaction time, the other is the interaction between adsorbents and adsorbates [36]. With the addition of adsorbent, the concentration of SMX decreased rapidly during the first 5 min, then the decrease gradually slowed down, before finally reaching equilibrium. BC exhibited the longest time of 48 h before arriving at equilibrium, while ClBC and PBC reached equilibrium within 24 h, indicating that the material reached equilibrium in a relatively short time due to the presence of sufficient adsorption sites at the beginning of the reaction, which was consistent with the surface area above. When fitting the data, two kinetics models were applied to describe the obtained data in order to explore the adsorption behavior of SMX on BCs. Usually, R^2^ was applied to estimate the suitability of different models; a higher value indicates a better fit [37]. As listed in Appendix A (refer to Appendix A), the three groups of data have a high degree of fit using the PSO model. Their R^2^ values were 0.888, 0.993 and 0.993, respectively. These results indicate that the sorption of SMX on sorbents was dominated by chemical sorption, similar to most studies on sulfamethoxazole adsorption [38,39].

To further understand the adsorption process, the experiments we developed allowed us to investigate whether the intra-particle diffusion had a controlling effect [40]. As shown in Figure 4 and Appendix A (refer to Appendix A), the adsorption rate was controlled by intra-particle diffusion. This was closely related to the abundant pore structure observed by SEM.

The adsorption kinetic data can be divided into three parts by using intra-particle diffusion model fitting. In the first portion, with increasing reaction time, the adsorption amounts of total SMX tended to increase continuously, showing that the SMX in the liquid body diffused rapidly and aggregated to the biochar surface through the boundary layer. In particular, the surface of the biochar contains abundant functional groups and more micropores, and micropore diffusion plays a major role. In the second portion, the trend line slope decreased, and SMX diffused slowly from the liquid film to the micropores. Physical adsorption occurred in the first and second stages [41]. However, in the final stage, the diffusion rate is very slow, due to the small size of the internal micropores, and chemical sorption occurs in this stage. The analysis above suggests that the PSO model further shows the rate-limiting step in the adsorption process, which is a joint action of physical adsorption and chemical adsorption [42,43]. In addition, the curve of the first stage fitted by the intra-particle diffusion model does not pass through the origin. This indicates that the adsorption process is complex and is controlled by several rate-limiting steps [44]. In other words, the adsorption rate is controlled by intra-particle diffusion, but there are other rate-limiting steps.

### 3.3. Adsorption Isotherms

The Langmuir, Freundlich and Temkin models were fitted to the experimental data, as they have been widely used to explore the reaction process and mechanism. As is shown in Figure 5 and Appendix A (refer to Appendix A), the adsorption isotherm data of SMX on BC was fitted well by Freundlich model, and the correlation coefficient (R^2^ = 0.981) was higher. This indicates that SMX adsorption on the BC samples proceeds via a monolayer adsorption. However, from the R^2^ value and the fitting curve, we found that the Langmuir adsorption isotherm model (R^2^ = 0.941 and 0.923) was better able to demonstrate the behavior of ClBC and PBC adsorption than the Freundlich isotherm model (R^2^ = 0.928 and 0.900), indicating that the Langmuir model could more accurately describe the adsorption by ClBC and PBC [45]. The maximum capacity (*q*_m_) of ClBC and PBC was 48.84 mg·g^−1^ and 51.65 mg·g^−1^, according to the Langmuir model, which was far greater than the adsorption capacity of BC (35.40 mg·g^−1^). This suggests that SMX adsorption on the BC samples proceeds via a monolayer adsorption. The value of n is the degree to which the adsorption capacity depends on the equilibrium concentration. Values of n greater than 1 indicate favorable adsorption [45]. In addition, the *k*_f_ values of ClBC and PBC were significantly greater than that of BC, suggesting that hydrochloric acid and phosphoric acid could improve the capacity of SMX on BCs [46]. Thus, the higher values of *q*_m_ of ClBC and PBC compared to BC might be explained by the enhanced interaction between biochars and SMX for the increasing of surface area, oxygen-containing functional groups, and so on. This is similar to findings from previous studies [29,47] that suggest that pore filling and hydrogen bonding play a role between SMX and the biochar’s surface.

### 3.4. Thermodynamic Analyses

The adsorption process is also accompanied by a change in reaction system energy; the impact of temperature was studied at three temperatures (288.15 K, 298.15 K and 308.15 K). The thermodynamic parameters for SMX sorption are presented in Appendix A (refer to Appendix A). With the increase in temperature from 288.15 K to 308.15 K, the sorption properties were enhanced, and the *k*_f_ values of BC, ClBC and PBC at 308.15 K were higher than those at 288.15 K and 298.15 K. These results are similar to those reported in previous studies, in which the sorption was enhanced with high temperature [43]. To further evaluate the sorption behavior, the thermodynamic parameter values were calculated, and these are listed in Appendix A (refer to Appendix A). With a temperature range from 288.15 to 308.15 K, the spontaneity of the adsorption system is characterized by Gibbs free energy (Δ*G* < *0*). In addition, with increasing temperature, Gibbs free energy decreases, and the degree of spontaneity also decreases. The fact that the Gibbs energy does not change much in terms of absolute value means that temperature is not the main factor in the reaction, which indicates that high temperature is more favorable for adsorption [48].

The resulting ΔH-values being greater than 0 proves that the adsorption reaction is an endothermic process, and that the increase in temperature is conducive to the forward direction of the adsorption reaction. The ΔH-values for the adsorption of the three materials were less than 84 kJ·mol^−1^, which could indicate that physical forces, including ion exchange and van der Waals forces, played a major controlling role in the sorption of SMX by sorbents [8]. Moreover, the positive value of ΔS demonstrated that system chaos increases during the adsorption process, and the solid–liquid interface is random, which could be caused by the adsorption of more water molecules at the solid–liquid interface. The results above were similar to those of Liang Fei [47].

### 3.5. Influence of pH_initial_ of The Solution

To better understand the adsorption mechanism, experiments on the effect of the initial pH of the solution on the adsorption equilibrium were also carried out. The zeta potentials of the biochars are shown in Figure 6A. It has been proved that the pH of the solution affects whether the surface of the adsorbent is positively or negatively charged [9]. It was noted that the material surface was positively charged, as the zeta potential was greater than zero. Conversely, the surface of the biochars was negatively charged. As illustrated in Figure 6B, the three biochars share similar patterns. The capacity firstly showed a rising trend, and then fell with increasing pH. It can be seen that the pH value had a significant effect on the sorbent adsorption of sulfamethoxazole. Additionally, the effect of pH value on the distribution coefficient (*K*_d_) in BC, ClBC and PBC exhibited the same trend. The form of SMX in solution had a significant effect on the properties of the adsorbent, too. The adsorption increased gently when the initial pH values were 5, 6, and 7 on BC, with a maximum *K*_d_ value for SMX at pH 7, followed by a very rapid decline with pH from 7 to 8. Finally, when the pH increased to 10, the capacity remained roughly consistent. In addition, as the pH increased from 2 to 5, the adsorption capacity of ClBC and PBC for SMX reached its maximum value. When the pH further increased from 5 to 8, there was a weak declining trend for adsorption capacity. As the pH is further increased from 8 to 10, the adsorption capacity remained in balance. This is inextricably related to the chemical form of the amphoteric compounds in the aqueous solution.

The surface charge of the adsorbent and the ionic morphology of the compound will change due to the presence of large numbers of hydrogen ions or hydroxide ions in the solution. Simultaneously, sulfamethoxazole is an amphoteric compound, and amino and sulfonate amines on the benzene ring of sulfamethoxazole can be ionized. The molecular morphologic distribution of sulfamethoxazole in water changes with pH. The solution exists as a complex mixture of cations, anions and neutral molecules. When the pH of the solution is in the range from 1.6 to 5.7, sulfamethoxazole mainly exists in the form of neutral molecular states without charge. With the increase of pH value, the proportion of neutral molecular morphology decreases, and the gradual increase of anion morphology increases the electrostatic repulsion between sulfamethoxazole and biochar, causing the gradual decrease of adsorption amount. In the pH range 3–7, there was little electrostatic interaction between the primary neutral species of SMX^0^ and the negatively charged adsorbent. The maximum adsorption capacity on BCs observed at pH 5 and pH 7 might be attributable to the π-π EDA interaction. When pH increased to 8, the content of SMX^0^ gradually decreased, and the content of SMX^−^ increased. At this time, the electrostatic repulsion between sulfamethoxazole and ClBC or PBC significantly hindered adsorption onto the materials. This suggests that electrostatic reactions play a role in it, but π-π EDA interaction was the predominant mechanism in this system [49]. In summary, the altered charges of both SMX and biochars could reasonably explain the changes of sorption capacity of SMX on biochars at different pH values. At low pH, SMX was cationic and the surface of biochars was commonly positive. Therefore, the π-π electron donor–acceptor interaction (π-π EDA) was weak. With increasing pH, SMX became zwitterionic, and the π-π EDA between SMX and biochars was strengthened. Thus, the enhanced sorption of SMX was observed [50].

### 3.6. Effect of Ionic Strength and Types

Because the strength of the ions strongly affects the activity coefficient of the compounds transferred from the solution to the surface, it was necessary to perform experiments adding different concentrations of NaCl to the mixture of BCs and SMX solutions. As shown in Figure 7A, the results revealed that Na^+^ was able to slightly increase the SMX adsorption at low concentrations (0.05 mol·L^−1^) after 72 h. It has been reported that the electrolyte in the solution affects the adsorption process because of electrostatic shielding. On the one hand, it can change the interaction intensity of adsorbents with adsorbents, and on the other hand, it can compete with antibiotics to adsorb sites on the adsorbents [51]. This discovery could be attributed to electrolytes, which are able to improve the adsorption capacity of sulfamethoxazole on the surface of biochars, because salting out causes an increase in the distribution of sulfamethoxazole on the biochar surface. However, with a further increase in concentration, the capacity of the sorbents decreased gradually, and then slowly approached equilibrium. This was probably due to the competition between Na^+^ and SMX to occupy active sites. The results suggest that electrostatic interaction might be the main mechanism [52].

The impacts of ionic types on the removal of sulfamethoxazole were surveyed in the presence of varying valences (including cations and anions) at a concentration of 0.1 mol·L^−1^. The results are shown in Figure 7B,C. Figure 7B indicates that, with the introduction of a large number of cations into the water, a competitive effect with sulfamethoxazole molecules is produced. The strength of the force mainly depends on the valence and radius of the ions. The larger the valence and radius of the ions, the stronger the competitiveness. In addition, although Al^3+^ have the highest valence state and ion radius, the effect on adsorption is not obvious due to hydrolysis. Figure 7C showed that sorption was influenced greatly by OH^−^, PO_4_^3−^, CO_3_^2−^, HCO_3_^−^. There might be two explanations for this phenomenon, one being that anions compete with sulfamethoxazole for adsorption sites, and another being the solution pH > 7 caused by hydrolysis of anions [53]. However, there were only relatively minor effects with respect to NO_3_^−^ and SO_4_^2−^ due to their neutral molecules.

### 3.7. Regenerability and Stability

The cyclic adsorption and regeneration properties of adsorbents are very important for their long-term operation. In practical applications, the recyclability of adsorbents is another key factor. The good regeneration performance of the adsorbent not only reduces secondary pollution, but also greatly reduces the cost of sewage treatment.

The regenerability of BC, ClBC and PBC were estimated by repeating the simulated adsorption–desorption process five times with 0.1 mol·L^−1^ NaOH as the desorption agent. As illustrated in Figure 8, after five cycles, the capacities of BC, ClBC and PBC all decreased. As for SMX, the equilibrium adsorption capacities of BC, ClBC and PBC decreased from 15.23 mg·L^−1^, 26.66 mg·L^−1^ and 26.77 mg·L^−1^ to 13.59 mg·L^−1^, 23.96 mg·L^−1^ and 24.32 mg·L^−1^. The equilibrium adsorption capacities of BC, ClBC and PBC only decreased by 10.77%, 10.13%, and 9.15%. This demonstrated that the modified biochars still retained their performance after multiple cycles, leading to the conclusion that alfalfa biochar has great regenerability.

### 3.8. Comparative Study

However, it is difficult to compare the capacity of various adsorbents directly, because of the differences in adsorbates. Table 1 demonstrates the superiority of biochar prepared from alfalfa biomass by comparing the adsorption capacities (Q_m_) of biochar from nine different sources reported in the literature for sulfamethoxazole. The Q_m_ of modified alfalfa biochar (51.65 mg·g^−1^) was higher than other materials such as Chinese medicine residues (Deng et al. 2020), spent coffee grounds (Zhang X et al. 2020), bagasse powder (Zhang R et al. 2020), giant reed (Zheng et al. 2013), hickory chips (Huang et al. 2020), pomelo peel (Cheng et al. 2020), rice straw (Han et al. 2013), pinus radiata sawdust (Reguyal et al. 2017) and sludge (Ma et al. 2020). Thus, alfalfa biochar was clustered as the starting materials for the removal of SMX.

## 4. Conclusions

The initial biochar was produced from alfalfa by pyrolysis at 800 °C and then etched with H_3_PO_4_ or HCl. With the acid modification, the surface area, aromaticity and oxygen-containing functional group increased greatly. By evaluating the adsorption onto biochars, the sorption affinity of the biochar to SMX followed the order PBC > ClBC > BC, while the adsorption capacity was more than three times greater than before. The higher sorption on BCs was imputed to its abundant microstructure and surface functional groups. After analysis, it was concluded that pore filling, electrostatic interaction and π-π EDA might be the main mechanisms in adsorption, while hydrophobicity might a certain role in it. The PSO model was able to well describe the adsorption kinetics on BCs. The adsorption data of SMX on ClBC and PBC fitted the Freundlich model better, except for SMX on BC. Thermodynamic analysis showed that the removal of sulfamethoxazole by biochar was a spontaneous endothermic process. These findings illustrate the great impact of acid etching on the interactions between ionizable pollutants and biochars with special physicochemical properties. This research demonstrates the promise of alfalfa as a novel efficient and environmentally friendly sorbent for the removal of sulfamethoxazole from aqueous solutions that is easy to obtain, cost effective, and convenient.

## Figures and Tables

**Figure 1 materials-14-01033-f001:**
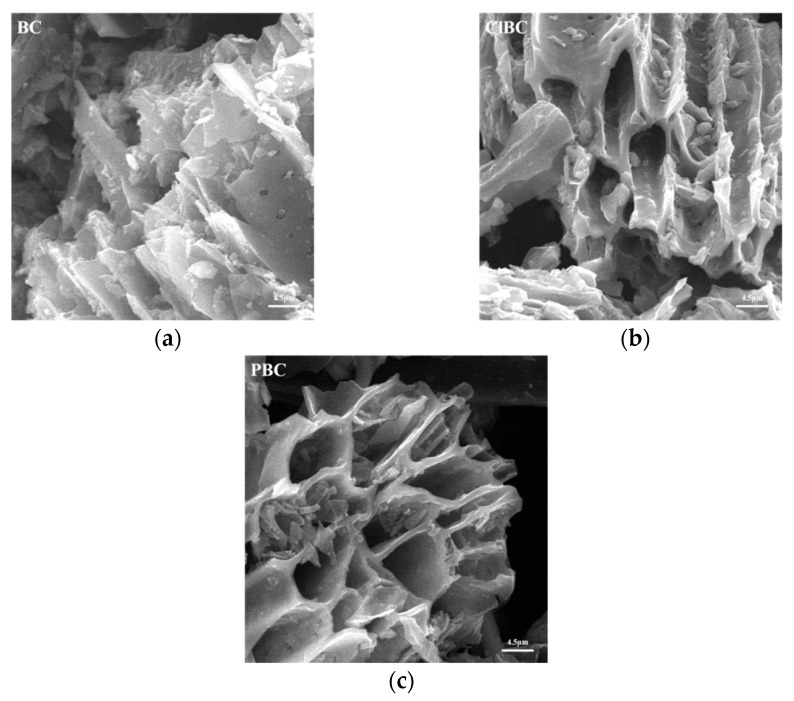
SEM images of modified biochars (**a**) raw bochar, (**b**) HCl-modified biochar and (**c**) H_3_PO_4_-modified biochar.

**Figure 2 materials-14-01033-f002:**
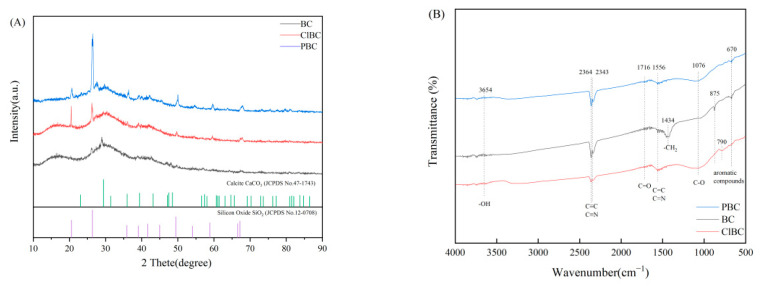
XRD patterns (**A**) and FT-IR spectra (**B**) of BC, ClBC and PBC

**Figure 3 materials-14-01033-f003:**
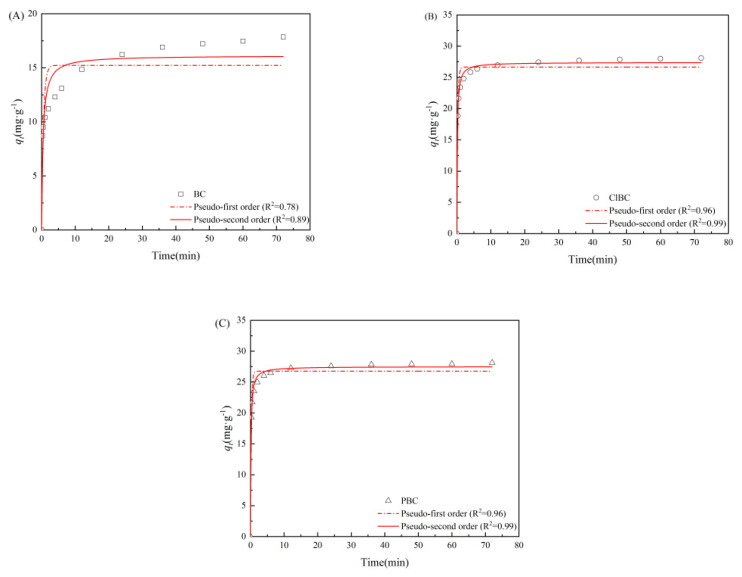
PFO (pseudo-first order) and PSO (pseudo-second order) model for SMX adsorption on (**A**) BC, (**B**) ClBC and (**C**) PBC.

**Figure 4 materials-14-01033-f004:**
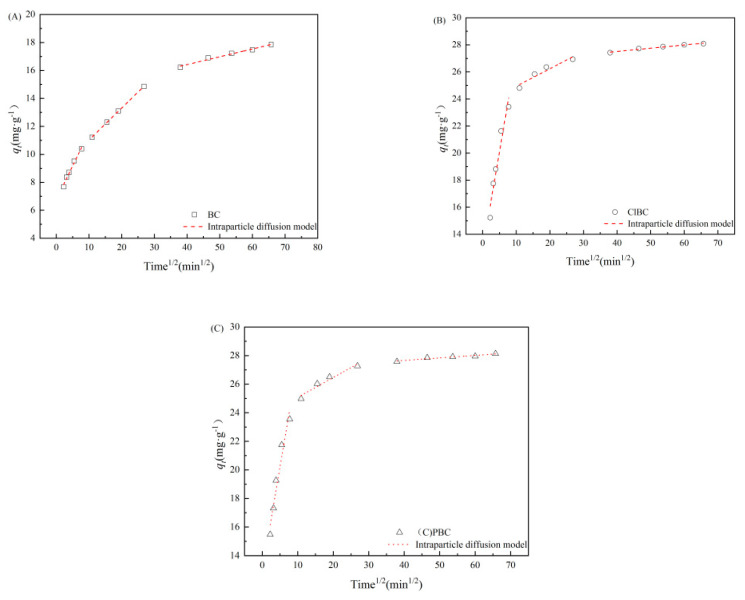
IPD model for SMX adsorption on (**A**) BC, (**B**) ClBC and (**C**) PBC.

**Figure 5 materials-14-01033-f005:**
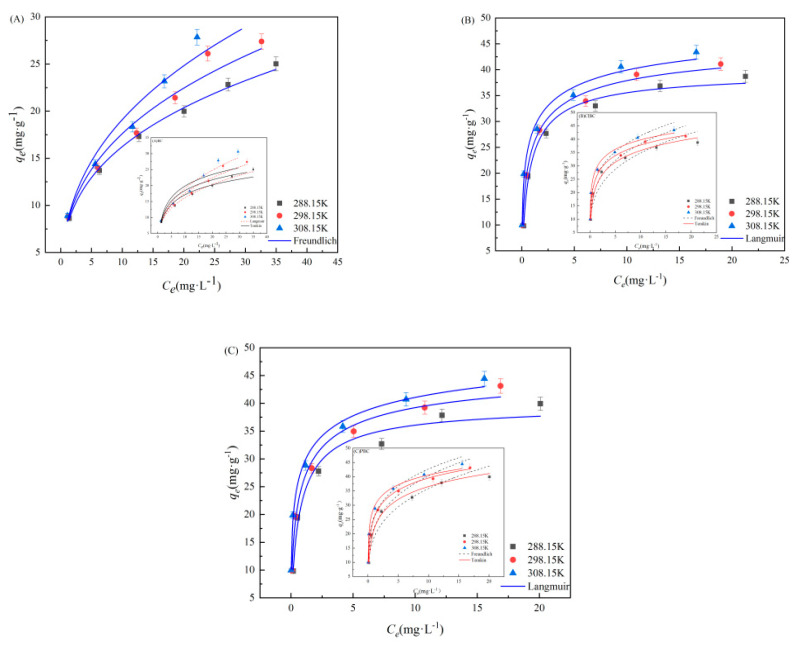
Langmuir, Freundlich and Temkin model for SMX adsorption on (**A**) BC, (**B**) ClBC and (**C**) PBC.

**Figure 6 materials-14-01033-f006:**
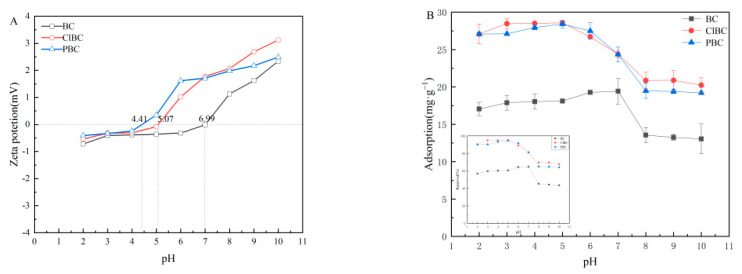
(**A**) Zeta potential of biochars; (**B**) influence of initial pH on adsorption.

**Figure 7 materials-14-01033-f007:**
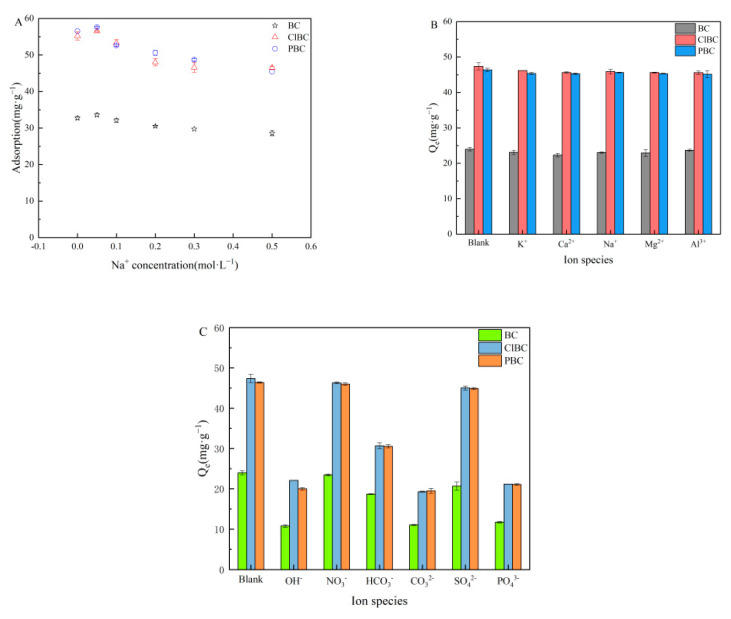
Effect of ionic strength and types. (**A**) ionic strength; (**B**) cations; (**C**) anions.

**Figure 8 materials-14-01033-f008:**
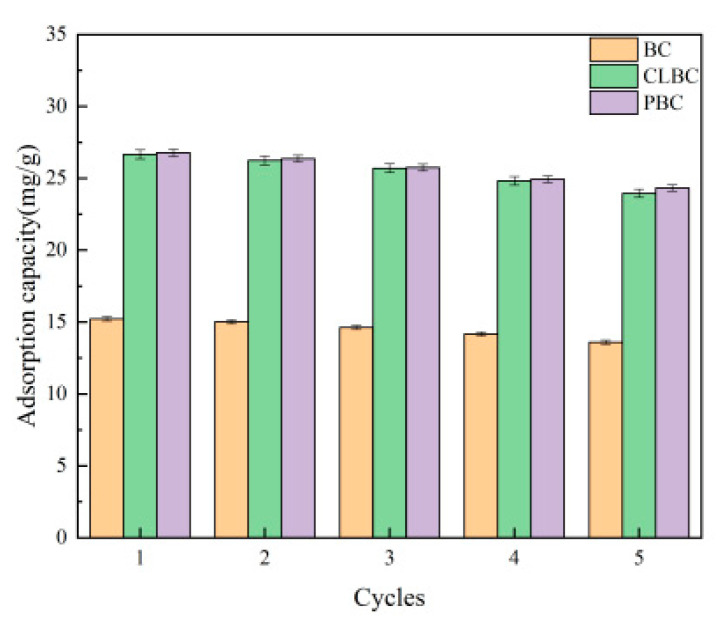
Adsorption–desorption cycles for SMX on BC, ClBC and PBC.

**Table 1 materials-14-01033-t001:** Comparison of the maximum adsorption capacities of SMX onto various sorbents.

Adsorbents	Modification Methods	Q_m_(mg·g^−1^)	References
Chinese Medicine Residues	nHAP@biochars	51.22	[54]
Spent Coffee Grounds	Pyrolysis at 450 °C	0.13	[55]
Bagasse Powder	Pyrolysis at 800 °C	35.43	[8]
Giant Reed	Pyrolysis at 300 °C	4.99	[56]
Hickory Chips	Ball Milled	24.30	[57]
Pomelo Peel	Modified by KOH and HNO_3_	0.832	[38]
Rice Straw	Pyrolysis at 400 °C	1.828	[58]
Pinus Radiata Sawdust	Modified by NaOH	17.49	[49]
Sludge	Modified by HCl	5.43	[59]
Alfalfa	Pyrolysis at 800 °C	35.40	Present Work
Alfalfa	Modified by HCl	48.85	
Alfalfa	Modified by H_3_PO_4_	51.65	

## Data Availability

The study did not report any data.

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
