# Peer review of "Sorption of Sulfamethoxazole on Inorganic Acid Solution-Etched Biochar Derived from Alfalfa"

_materials, 2021, doi:10.3390/ma14041033_

Round 1
Reviewer 1 Report
The manuscript reports the removal of antibiotic sulfamethoxazole using alfalfa biochar as sorbents. Both the physical-chemical characterization of the sorbents and the adsorption study have been well performed and presented. However, the manuscript could be accepted for publication after minor revision, and the following concerns need to be addressed:
- The sorbents used in this manuscript should be compared with similar ones reported in the literature.
- The acronyms SMX, ClBC and PBC have been used in the abstract. However, their meaning is clarified in materials and methods. Authors should define the abbreviations the first time they appear in the text.
- The sentence between 62-68 lines is not very clear. Authors are advised to check this part.
- In paragraph 2.2, it is suggested to describe in detail the three different sorbents BC, PBC and ClBC.
- It would be advisable to cite the opportune bibliographic references about the kinetic equations used in the paper.
- About SEM measurements, same dimensional values of pores should be added to comments to compare the different sorbents. In addition, it is suggested to report the size bar on the SEM images.
- To have very efficient materials for adsorption processes with a good recyclability, sorbents should also be reused in subsequent adsorption cycles. The authors believe that the studied materials can carry out desorption and subsequent re-adsorption processes of antibiotic?
Reviewer 2 Report
The authors prepared alfalfa-derived biochar treated with inorganic acids for the removal of the pollutant sulfamethoxazole. They presented promising results however, characterization of the materials seems a bit lacking such as when the surface area and other textural characteristics were merely presented in Table 1 without further explanation on what method or equation was used to estimate or compute them. Results were also not thoroughly elaborated and need further explanation to prove their points and conclusion. The paper also needs extensive English editing as it contained a lot of misspelled words and grammatical errors.
Comments:
- The title makes the research seem like the biochar contains sulfamethoxazole that needs to be removed. Please revise the title accordingly.
- It was mentioned in the introduction that “alfalfa meal is one of the best sources for protein, minerals and cellulose, and it is a rich source of minerals as well as vitamins”. As alfalfa has a lot of potential to be used productively, why did you consider it as the source for the biochar instead of other bio-waste that have no other good use instead?
- Kindly improve on the labels on the SEM images on Figure 1 as they are not quite readable.
- Include footnotes for Table 1 to interpret the symbols used (Vtand Dm).
- The maximum adsorption capacity of the prepared material was 51.65 mg/g which seems promising. Kindly provide a table to summarize the recent works on sulfamethoxazole adsorption with their corresponding maximum adsorption capacity to highlight the advantage of your work over previous researches/literature.
- Page 7, Line267 “SMX in the liquid body diffused rapidly and aggregated to the biochar surface”; Page 9, Line 306 “suggests that pore filling and 306 hydrogen bonding played a role between SMX and the biochar’s surface”; Page 12, Line 365 “π-π EDA interaction was the predominant mechanism 365 between SMX and the biochar’s surface in single system”. With these statements, the authors suggest the SMX is adsorbed on the biochar’s surface. Was surface area analysis conducted once again after the SMX adsorption on biochar?
- Page 11, Line 336: “The Zeta potential of biochars were shown in Figure. 6(A). It has been proved that the surface charge of sorbent varies with changing in the pH of solution[9].” – No further discussion on the effect of the pH to zeta potential, to the adsorption was mentioned. Please substantiate.
- In Figure 6B, it can be observed that the trend for SMX adsorption on BC is upwards from pH 5-7. However, for C1BC and PBC, it is trending downwards. The short explanation in page 11 line 361 onwards is quite lacking. Please elaborate more regarding this difference.
Round 2
Reviewer 1 Report
I thank the authors for taking into consideration my comments to improve the quality of the paper and the authors' answers are exhaustive.
Moreover, the new title is much clearer than the previous one.
In my opinion the new revised version of the manuscript could be accepted for publication.
Best regards.
Reviewer 2 Report
The authors have addressed the comments and the paper can be accepted now.